# Levels of Impulsivity, Hyperactivity, and Inattention and the Association with Mental Health and Substance Use Severity in Opioid-Dependent Patients Seeking Treatment with Extended-Release Naltrexone

**DOI:** 10.3390/jcm10194558

**Published:** 2021-09-30

**Authors:** Ann Tarja Karlsson, John-Kåre Vederhus, Thomas Clausen, Bente Weimand, Kristin Klemmetsby Solli, Lars Tanum

**Affiliations:** 1Addiction Unit, Sørlandet Hospital HF, 4604 Kristiansand, Norway; john-kare.vederhus@sshf.no (J.-K.V.); thomas.clausen@medisin.uio.no (T.C.); 2Norwegian Centre for Addiction Research, University of Oslo, 0315 Oslo, Norway; k.k.solli@medisin.uio.no; 3Mental Health Services, Akershus University Hospital, Lørenskog, 1478 Oslo, Norway; bente.weimand@usn.no; 4Center for Mental Health and Substance Abuse, University of South-Eastern Norway, 3040 Drammen, Norway; 5Vestfold Hospital Trust, 3103 Toensberg, Norway; 6Department of R&D in Psychiatric Health Care, Akershus University Hospital, 1478 Oslo, Norway; lars.hakon.reiestad.tanum@ahus.no; 7Faculty for Health Science, Oslo Metropolitan University, 0130 Oslo, Norway

**Keywords:** extended-release naltrexone, opioid dependence, mental distress, impulsivity

## Abstract

The level of impulsivity, hyperactivity, and inattention (IHI) is higher among patients with substance use disorder (SUD) than in the general population. However, the prevalence of such symptoms in patients seeking treatment with an opioid antagonist, such as extended-release naltrexone (XR-NTX), is unknown. We screened 162 patients with opioid use disorder (OUD) seeking treatment with XR-NTX in Norway using the Adult ADHD Self-Report Scale (ASRS) to estimate the prevalence of IHI alongside an assessment of mental and physical health and substance use. Sixty-six patients scored above the clinical cut-off on the ASRS. Higher levels of IHI were significantly associated with a longer history of frequent amphetamine use, current alcohol use, and greater mental distress. Mental distress was the strongest factor associated with higher levels of IHI. The introduction of screening for IHI and mental distress in opioid maintenance treatment and XR-NTX would likely improve the quality of care and enable clinicians to tailor interventions to the needs of patients with high levels of IHI to prevent treatment discontinuation.

## 1. Introduction

A high prevalence of psychiatric comorbidity, mental distress, and attention deficit hyperactivity disorder (ADHD) has been observed in a number of patient populations with substance use disorder (SUD) [1,2]. However, knowledge of the impact of ADHD symptomatology as a co-occurring condition among opioid-dependent patients enrolled in opioid maintenance treatment (OMT) is sparse [3,4]. The core symptoms of ADHD are impulsivity, hyperactivity, and inattention (IHI) and difficulties performing executive functions. In the treatment of SUDs, these are known risk factors for decreased adherence and an escalated risk of discontinuation [5,6]. Previous findings have shown that the presence of IHI is associated with earlier onset and a more severe course of illicit substance use [7]. Diagnosing ADHD is a comprehensive and challenging task, even more so when co-occurring with SUD [8]. However, IHI can be symptoms of many conditions and not solely represent ADHD. Therefore, independent of diagnosis, IHI may represent an obstacle in the treatment and recovery of patients in OMT programs.

By 2019, there were approximately 7900 patients enrolled in Norwegian OMT programs, and the primary medical treatment option was opioid agonist therapy, such as methadone, buprenorphine, or buprenorphine/naloxone [9]. These are powerful prescription medications that relieve the need for illicit drugs, but they also represent a continued need for opioids and, thus, maintain opioid tolerance. Diversion of such medication is a known and undesired side effect of OMT at a societal level [10].

Medical treatment with an extended-release opioid antagonist is an alternative to OMT medication for select patients and may reduce some of the concerns associated with OMT [11]. However, premature discontinuation of antagonist treatment could expose patients to an increased risk of relapse, and even opioid overdose due to a reversal of their opioid tolerance [12].

To explore the burden of IHI, it is relevant to better understand the impact of these risk factors on treatment trajectories [13]. To the best of our knowledge, no previous study has explored IHI in patients with opioid use disorder (OUD) preferring opioid antagonist to opioid agonist treatment. This would allow us to elaborate on the future perspectives of additional treatment needs for patients with increased levels of IHI. The findings will show whether the prevalence of IHI is higher among those who seek out this new and novel treatment option compared to other relevant studies in the SUD treatment field. Such findings may indicate a general preference for novelty and new experiences among patients with higher IHI scores.

We estimated the prevalence of patients scoring above the clinical cut-off on a screening test for IHI in a population qualifying for OMT, but with a preference for opioid antagonist treatment. To broaden our understanding of the challenges encountered by these patients, particularly in treatment settings in which non-adherence to treatment incurs increased risks, we examined the reported level of IHI and its association with SUD complexity and mental health burden. Lastly, we explored which factors had the strongest association with elevated levels of IHI.

## 2. Materials and Methods

This study was built upon baseline data from the Norwegian NaltRec study (“Long acting naltrexone for opioid addiction: the importance of mental, physical, and societal factors for sustained abstinence and recovery”) [14], an open label multi-centre study carried out at five urban addiction clinics across Southern Norway (Akershus University Hospital, Sørlandet Hospital, Vestfold Hospital, Oslo University Hospital, and Haukeland University Hospital). It is a phase IV study on extended-release naltrexone (XR-NTX) in the treatment and recovery of patients with opioid dependence qualifying for OMT in a naturalistic setting. After complete detoxification from all opioids, an injectable suspension of 380 mg of extended-release naltrexone hydrochloride (Vivitrol^®^) was administered every 4 weeks. The total study period was 24 weeks, followed by a 28-week optional follow-up treatment period. Study interviews, assessments of mental distress, and urine drug screens were performed at each visit together with the XR-NTX injection.

The study was conducted in accordance with the ethical principles of the Declaration of Helsinki [15], which are consistent with the International Conference on Harmonisation (ICH) guidelines for Good Clinical Practice (GCP) [16] and national regulatory requirements. Patient data were recorded and handled in accordance with the General Data Protection Regulation (GDPR) and the National Personal Data Protection regulations. All patients provided informed consent before the start of the study. All patients were enrolled or continued in an OMT program at inclusion to secure adequate follow-up and availability of opioid agonist therapy in case of early discontinuation of XR-NTX treatment. Co-researchers from user organizations took part in the development of the study. The study is registered at clinicaltrials.gov (NCT01717963).

### 2.1. Setting and Patients

Patients were recruited between September 2018 and September 2020 from addiction clinics and detoxification units at the participating hospitals. A number of patients were also referred from the municipal health services and by word of mouth between opioid users. Men and women aged 18–65 years with a current diagnosis of opioid dependence were eligible for participation [17]. Patients had to be capable of understanding the implications of treatment and be willing to comply with the study procedures. If not already enrolled in an OMT program, patients were obligated to enrol at study inclusion. Patients with severe psychiatric or physical illness that demanded treatment that could interfere with study participation were excluded from participating. In addition, pregnant and lactating women and people with a primary alcohol dependence were excluded from the study. Women of childbearing age were constrained to use a safe contraceptive of choice.

### 2.2. Measures

All screening procedures were carried out by trained staff at the participating hospitals. Demographic data were obtained using the European version of the Addiction Severity Index interview (Europ-ASI). Opioid dependence was verified by The Mini International Neuropsychiatric Interview (MINI), version 6.0 [18].

IHI was measured using the self-administered Adult ADHD Self-Report Scale 18-item version (ASRS-18) v1.1 [19]. The ASRS-18 consists of a combined impulsivity and hyperactivity sub-scale and an inattention sub-scale with nine questions in each. The questions assessed how often a symptom occurred over the past 6 months on a 0–4 scale with responses of never (score = 0), rarely (1), sometimes (2), often (3), and very often (4). We used the optimal scoring procedure described by Kessler et al. [19] with dichotomization of each question [19]. Clinically significant symptom levels were defined for seven of the questions as responses of sometimes, often, and very often (e.g., “How often do you have problems remembering appointments or obligations?”), and for the remaining 11 questions, clinically significant symptom levels were defined as often and very often (e.g., “How often do you feel restless or fidgety?”). A summed score was calculated (range 0–18) and then dichotomized; a score ≥ 9 indicated an overall clinical symptom level of IHI. We used both the summed and the dichotomized score in the analyses. In the original study, the sensitivity of the scoring method was 56.3% and the specificity 98.3% [19]. For comparison, we also report findings from the 6-item version of the ASRS (ASRS-6). This shorter version of the scale includes four inattention items and two hyperactivity items (i.e., no items pertaining to impulsivity). It was recommended by Kessler et al. [19] as an initial screening tool for ADHD and has been used, to some extent, in populations with SUDs [19,20,21]. The scoring method is similar to that of the full version, with the summed score ranging from 0–6, and a score ≥ 4 is considered to be a likely positive screen for ADHD [3,4,22]. Compared to the 18-item version, which was validated in a representative community sample, Daigree et al. [20] reported a much higher sensitivity (87.5%) and much lower specificity (68.6%) for the 6-item scale in a population with SUDs [20].

Current mental distress was measured using the 25-item Hopkin’s Symptom Checklist (H-SCL-25) [23]. This self-administered inventory measures symptoms of anxiety and depression in the past 14 days on a 4-point Likert scale ranging from 1 (“not at all”) to 4 (“extremely”). A mean score of all items is referred to as the Global Severity Index (GSI), and a score of 1.75 was set as the cut-off, with higher scores considered to indicate clinical levels of mental distress [23]. Historical mental health problems were assessed by the Europ-ASI asking whether patients had experienced episodes of serious mental health issues during their lifetime (e.g., depression, anxiety, cognitive challenges, psychosis, violent or suicidal behaviour).

Current severity in other life areas (i.e., somatics, employment, alcohol, drugs, legal issues, family) was measured using the Europ-ASI [24,25]. Composite scores were calculated for each area to indicate severity during the past 28 days and ranged from 0 (no problem) to 1 (a severe problem). Historical severity of drug use was measured using the Europ-ASI and assessed as the duration of high frequent use of these drugs.

### 2.3. Statistical Analysis

We used descriptive statistics to characterize the sample. Some data were not normally distributed and were reported as medians and interquartile ranges (IQRs).

Non-parametric statistics (Mann–Whitney U-test) were applied to compare the groups in regard to continuous variables and cross-table analysis with the chi-squared test for categorical variables. We used linear regression to examine the association between the IHI score and independent variables, such as socio-demographic characteristics and severity variables. The results of these analyses are presented as unstandardized beta coefficients (*β*) with a 95% confidence interval (CI). Preliminary bivariate analyses were performed and variables with significant p-values were included in a multivariable linear regression analysis. The R-squared (R^2^) value was used to assess the percentage of variation in the level of IHI explained by the model. We also report the standardized βs to compare the relative strength of the relationship between variables. The threshold for significance was set at *p* < 0.05. All statistical analyses were performed using IBM SPSS, version 26 [26].

## 3. Results

The sociodemographic variables of the 162 patients who participated in this study are presented in Table 1 according to ASRS-18 scores above and below the cut-off. Most participants were male, more than half of the patients were living alone, and most had never been married. In terms of IHI, 66 (41%) patients scored above the clinical cut off on the ASRS-18. There were no between group differences, except for a lower number of completed years of education among the patients who scored above the cut-off on the ASRS-18. Using the alternative short version of the scale (ASRS-6), the number of patients scoring above the cut-off was somewhat higher (47%).

In terms of the severity of past drug use, the patients who scored above the cut-off on the ASRS-18 reported significantly longer histories of frequent use of alcohol and amphetamines (Table 2). Among the patients who scored above the cut-off on the ASRS-18, there was a larger proportion of self-reported severe symptoms on the Europ-ASI regarding cognitive challenges (perception, concentration, and memory), non-drug-related hallucinations, violent behaviour, and suicidal behaviour during the participant’s lifetime.

Looking at current severity, the patients with ASRS-18 scores above the cut-off had significantly more somatic symptoms measured using the Europ-ASI composite scores. Scores on the H-SCL-25 were also significantly higher in the patients with an ASRS-18 score above the cut-off, indicating a higher level of mental distress during the 2 weeks prior to screening (Table 2).

A multivariable regression analysis was conducted with past and current areas of severity that were significant in the bivariate analysis (Table 3). Of the sociodemographic variables, only the number of years of completed education was significant in the bivariate analysis and included in the regression analysis. The full model explained 31% of the variance in ASRS-18. Three of the variables were significantly associated with ASRS-18, they are as follows: composite score alcohol, years of frequent use of amphetamines, and mental distress measured using the H-SCL-25 (Table 3). The standardized β values were 0.15 for composite score alcohol, 0.22 for years of frequent use of amphetamines, and 0.27 for mental distress. Thus, current mental distress showed the strongest association with a higher level of IHI.

## 4. Discussion

In this study, the patients scoring above the ASRS-18 cut-off reported longer histories of frequent amphetamine and alcohol use, as well as more past and current psychiatric symptoms. A longer history of frequent amphetamine use and higher current mental distress had the strongest association with the level of IHI.

Previous studies in the field of SUD treatment mainly used the 6-item version of the ASRS [3,4,20]. Therefore, we used results from this shorter version to compare the prevalence with other studies. The proportion of patients with ASRS-6 scores above the cut-off was higher in our study than in a previous Norwegian study (47% vs. 33%, respectively) [3]. Our proportion of patients above the cut-off was also slightly higher than in the International ADHD in Substance Use Disorders Prevalence Study (IASP), in which the overall prevalence was 40.9% (vs. 47%) [21]. This indicates a higher proportion of patients scoring above the cut-off on the ASRS-6 in our study population than previously reported in OMT program populations. In the present study, the prevalence of patients scoring above the cut-off was somewhat lower on the 18-item version, indicating that the 6-item version is a more sensitive measure in populations with SUDs. Notably, a positive ASRS score in samples with SUDs cannot be interpreted as a likely ADHD diagnosis, as Kessler et al. [19] recommended when the ASRS was used in a general population sample [19]. In the IASP study, approximately one-third of positive ASRS-6 scores were confirmed as cases with ADHD when diagnostic interviews were carried out [21]. Thus, ASRS screening above a clinical cut-off should only be considered as a first step when aiming to identify ADHD.

The patients with ASRS-18 scores above the clinical cut-off presented longer histories of stimulant drug use (amphetamines). This is in line with previous studies indicating a preference for stimulants among the patients with symptoms of IHI [3,22]. Comparably, a recent Australian study found that amphetamine use as a primary substance of concern was a negative predictor of treatment completion [27].

Looking at the lifetime prevalence of psychiatric symptoms, a larger fraction of the patients with ASRS-18 scores above the cut-off reported cognitive challenges (perception, concentration, and memory), non-drug-related hallucinations, violent behaviour, and suicidal behaviour. The same applied to current mental distress, as a significantly higher proportion of those scoring above the ASRS-18 cut-off reported clinical levels on the H-SCL-25 (70% vs. 48%). Similar findings have been reported in previous studies of patients with SUDs, as well as in patients in OMT [1,28].

In a multivariable analysis, high levels of mental distress were more strongly associated with IHI than other variables. One increased point on the H-SCL-25 was associated with a 2.29-point increase on the ASRS 18-item scale when controlling for all the other factors in the model. This is a substantial increase and implies a need to focus on current mental distress when considering feasible treatment options for patients with OUD. A previous randomized study conducted in Norway found that patients receiving opioid antagonist treatment did not experience any increase in mental distress over a 3-month period compared to patients on BUP/NLX [29]. In the subsequent 6-months follow-up period, the study found a slight decrease in mental distress over time, indicating no adverse effect of XR-NTX on mental distress. Therefore, patients should not be discouraged from choosing an antagonist treatment over agonist treatment out of concern for a potential worsening of symptoms of depression and anxiety. Yet, the high levels of mental distress among the participants in our study calls for clinical attention. The proportion of patients with mental distress above the clinical cut-off was especially prominent (70%) within the group of patients with contemporarily high levels of IHI at treatment initiation. Taking into consideration that many of these patients carry a double burden of elevated levels of mental distress and increased IHI, clinicians should bear in mind the challenges these patients face when entering treatment. Moreover, higher levels of mental distress were associated with an increased risk of discontinuing treatment in a recent Norwegian study [30]. Thus, as part of a screening procedure prior to the induction of XR-NTX, questionnaires tailored to identifying these types of challenges would enable clinicians to better meet the needs of patients who are extra vulnerable to relapse due to increased mental distress combined with poorer inhibitory control.

Regarding IHI, the ASRS-18 does not provide information on any threshold for neurocognitive impairment or the symptom aetiology. It does, however, hint to a state of executive functioning that is prone to being challenged in the process of recovering from SUD, taking into consideration the effort needed to resist relapse [31]. Patients with combined SUD and increased IHI would benefit from treatment focusing on aspects of cognitive functioning [32]. Reducing bottom-up processing in which decisions are based primarily on impulsivity and immediate dopamine reward in the subcortical brain regions could help prevent relapse [5]. A previous study showed that improving working memory could improve impulse control in methamphetamine users [32]. Furthermore, relieving patients from gaining an effect from, and to some extent reducing cravings for, opioids with XR-NTX treatment [11] could possibly create psychological safety and cognitive relief, thereby enhancing self-control and reducing impulsive behaviour.

### Methodological Considerations

The cross-sectional design limits the possibility of drawing conclusions about causality concerning the relationship between variables. The study could potentially have been strengthened by an in-study control group, such as a general SUD sample, to compare the prevalence of high ASRS scores. However, we have related our findings to comparable studies in this field. The IHI scores rely on subjective data, as the ASRS-18 is a questionnaire expressing the patient’s experience with IHI. No neuropsychological tests were applied to obtain more objective measures of IHI. Mental distress measured using the H-SCL-25 focused mainly on symptoms experienced during the past 2 weeks. At the time of screening and treatment initiation, the patients could have been influenced by residual withdrawal symptoms from drugs or excitement over entering a new treatment, influencing the H-SCL-25 scores [33]. During screening, a clinical evaluation was carried out by trained staff in order to avoid capturing states of ongoing withdrawal as measures of habitual states. In this way, we aimed to minimize confusing withdrawal with symptoms of IHI.

## 5. Conclusions

The high coexistence of IHI and mental distress warrants attention to improve the chance of successful treatment trajectories. With knowledge of this double burden among opioid-dependent patients seeking treatment with XR-NTX, a screening routine could help clinicians identify patients with an extended need for additional follow-ups in order to better facilitate the patient’s recovery process.

## Figures and Tables

**Table 1 jcm-10-04558-t001:** Sociodemographic characteristics of participants at baseline.

Variable	ASRS < 9	ASRS ≥ 9 (*N* = 66)	*p*-Value	Total (*N* = 162)
	*N* = 96	*N* = 66		
Female gender	21 (22%)	18 (27%)	0.430	39 (24%)
Age, years	38.4 (10.1)	37.1 (9.6)	0.474	37.9 (9.9)
Living conditions past 6 months				
- Living alone	55 (57%)	33 (50%)		88 (54%)
- With partner and/or children	18 (19%)	17 (26%)	0.721	35 (22%)
- With parents/other family/friends	12 (13%)	9 (14%)		21 (13%)
- Prison/institution/unstable housing	11 (12%)	7 (11%)		18 (11%)
Civil status				
- Married	5 (5%)	1 (2%)		6 (7%)
- Divorced/separated	14 (15%)	11 (17%)	0.326	25 (16%)
- Never married	77 (80%)	54 (82%)		131 (81%)
Not in OMT before enrolment in study	41 (43%)	20 (30%)	0.109	61 (38%)
Years of completed education	12.2 (2.5)	11.4 (2.5)	0.024	11.9 (2.5)

Values are reported as n (%) or mean (SD). *p*-values for group differences were obtained using Mann–Whitney U test or using chi-square. ASRS = Adult ADHD Self-Report Scale 18-item version; OMT = opioid maintenance treatment.

**Table 2 jcm-10-04558-t002:** Historical and current severity—substance use and mental health problems.

Frequent Use (Years) of	ASRS < 9 (*N* = 96)	ASRS ≥ 9 (*N* = 66)	*p*-Value
Alcohol (≥5 standard units per day) of alcohol per day	0 (4)	3 (30)	0.010
Heroin	4 (7)	5 (8)	0.217
Amphetamines	2 (6)	7 (12)	<0.001
Buprenorphine/methadone	4 (8)	5 (8)	0.247
Benzodiazepines	3 (8)	5 (11)	0.314
Cocaine	0 (0)	0 (2)	0.314
Other opioids	0 (2)	0 (2)	0.611
Cannabis	10 (18)	9 (14)	0.913
Historical severity—self-reported lifetime mental health problems measured by the Europ-ASI			
Depression	73 (76 %)	57 (86%)	0.105
Anxiety	82 (85 %)	61 (92%)	0.173
Cognitive challenges	70 (73%)	57 (86%)	0.041
Hallucinations	16 (17%)	21 (32%)	0.024
Violent behaviour	26 (27%)	31 (47%)	0.009
Suicidal behaviour	32 (33%)	33 (50%)	0.033
Current severity—Europ-ASI composite scores and mental distress			
Somatics	0.01 (0.50)	0.42 (0.83)	0.008
Employment	1.00 (0.33)	1.00 (0.33)	0.525
Alcohol use	0.01 (0.07)	0.04 (0.09)	0.067
Drug use	0.33 (0.31)	0.33 (0.32)	0.245
Legal issues	0.00 (0.20)	0.00 (0.20)	0.995
Family	0.03 (0.15)	0.03 (0.20)	0.923
H-SCL-25 GSI	1.72 (0.72)	2.04 (0.96)	<0.001
H-SCL-25 GSI > cut-off	46 (48%)	46 (70%)	0.006

Values are reported as median and interquartile range or n (%). *p*-values were obtained using the Mann–Whitney U test or using chi-square. ASRS = Adult ADHD Self-Report Scale 18-item version; Europ-ASI = European version of the Addiction Severity Index; H-SCL-25 GSI = Hopkin’s Symptom Checklist Global Severity Index.

**Table 3 jcm-10-04558-t003:** Factors associated with levels of inattention, hyperactivity, and impulsivity (IHI) (*N* = 162).

Variable	*β*	95% Confidence Interval	*p*-Value
Age	−0.06	−0.13–0.02	0.129
Gender	−0.49	−2.10–1.12	0.552
Years of completed education	−0.11	−0.41–0.19	0.478
Depression, lifetime	0.90	−0.97–2.77	0.342
Anxiety, lifetime	−0.13	−3.47–2.22	0.916
Cognitive challenges, lifetime	1.41	−0.36–3.19	0.117
Violent behaviour, lifetime	1.73	0.27–3.20	0.021
Suicidal behaviour, lifetime	0.39	−1.07–1.85	0.598
Composite score somatics	0.26	−1.69–2.20	0.795
Composite score employment	0.08	−2.69–2.86	0.952
Composite score alcohol use	9.03	0.33–17.73	0.042
Years of frequent amphetamine use	0.12	0.02–0.23	0.017
H-SCL-25 GSI	2.29	0.99–3.59	<0.001

Multivariable linear regression based on significant variables from the bivariate analyses. H-SCL-25 GSI = Hopkin’s Symptom Checklist Global Severity Index.

## Data Availability

The data used in this study were based on a still ongoing study that will be finalized in 2025. According to current Norwegian regulations and practice, the data will then be anonymized and deposited in a publicly available data repository (e.g., The Norwegian Centre for Research Data).

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
