# Peer review of "Levels of Impulsivity, Hyperactivity, and Inattention and the Association with Mental Health and Substance Use Severity in Opioid-Dependent Patients Seeking Treatment with Extended-Release Naltrexone"

_jcm, 2021, doi:10.3390/jcm10194558_

Round 1
Reviewer 1 Report
Authors were responsive to my comments.
Author Response
Reviewer 1:
Comments and Suggestions for Authors:
Authors were responsive to my comments.
Thank you for this acknowledgement.
Reviewer 2 Report
This significant paper examines the association between the major symptoms of ADHD and various mental health variables in patients with OUD seeking treatment with XR-NTX. Accumulating such research will help identify the difficulties of patients with OUD and lead to the development of more appropriate support methods.
I think that the introduction and discussion were well written, but I think that the analysis part was a little insufficient.
< Major comments >
1. In Setting and patients, please describe the number of subjects, gender ratio, mean age and standard deviation of age before dividing into ASRS <9 and ASRS ≥9.
2. Please combine Table1 to Table4 into one Table and add “Total” to Table2 to Table4. Also, for Table 1, please perform the test of the difference between ASRS <9 and ASRS ≥9 in the same way as Table 2 to Table 4. For Tables 1 to 4, please calculate the effect size.
3. Table 5 shows the results of crude that does not include covariates, but I think that the analysis results that consider the effects of confounding factors such as gender and age should also be shown in the paper. For example, it is highly recommended to perform an analysis using the sociodemographic characteristics of participants shown in Table 1 as covariates. Please revise Discussion based on the analysis result using covariates.
I hope that the above comments and suggestions may be helpful for you.
Author Response
Reviewer 2:
Comments and Suggestions for Authors:
This significant paper examines the association between the major symptoms of ADHD and various mental health variables in patients with OUD seeking treatment with XR-NTX. Accumulating such research will help identify the difficulties of patients with OUD and lead to the development of more appropriate support methods.
I think that the introduction and discussion were well written, but I think that the analysis part was a little insufficient.
< Major comments >
- In Setting and patients, please describe the number of subjects, gender ratio, mean age and standard deviation of age before dividing into ASRS <9 and ASRS ≥9.
The requested sociodemographic data were presented in table 1 and commented on in the beginning of the result section. Since this was a cross sectional study, we believe it is quite customary to place this information in the result section.
- Please combine Table1 to Table4 into one Table and add “Total” to Table2 to Table4. Also, for Table 1, please perform the test of the difference between ASRS <9 and ASRS ≥9 in the same way as Table 2 to Table 4. For Tables 1 to 4, please calculate the effect size.
Thank you for these suggestions. We consider that combining table 1 to 4 would be a matter of preference. Nonetheless, we have combined Table 2 – 4 as suggested. For simplicity, we did not add a total column to this rather large table.
We have included p-values in table 1 as suggested. Note that the regression analysis presented in Table 5 also included sociodemographic variables that were significant in bivariate analyses.
Since this is not a controlled study comparing different types of intervention, we reckon that it should be unnecessary to include effect size in table 1 and 2. Note that we reported a measure of effect size in the regression analysis (i.e., r2 value). The r2 value was >0.3, which is a quite large explained variance for the model.
- Table 5 shows the results of crude that does not include covariates, but I think that the analysis results that consider the effects of confounding factors such as gender and age should also be shown in the paper. For example, it is highly recommended to perform an analysis using the sociodemographic characteristics of participants shown in Table 1 as covariates. Please revise Discussion based on the analysis result using covariates.
Table 5 is a multivariable analysis based on significant variables in the bivariate analysis, and thus the results are reported as adjusted betas. We agree however, that it is common to include age and gender in this type of analysis although these variables were not significant in the bivariate analyses. Thus, we have now added age and gender to table 5 in the manuscript, as suggested. The manuscript text has been changed accordingly (one of the previous significant variables turned non-significant).
I hope that the above comments and suggestions may be helpful for you.
Thank you for the suggestions.
This manuscript is a resubmission of an earlier submission. The following is a list of the peer review reports and author responses from that submission.
Round 1
Reviewer 1 Report
The study investigated symptoms of hyperactivity, inattention and impulsivity in a treatment seeking sample of N = 162 opioid dependent patients in a cross-sectional design. The symptoms were assessed via a questionnaire on ADHD symptoms and compared to a few baseline characteristics (mainly a questionnaire on mental distress).
The manuscript is well written and the general topic (impulsivity / hyperactivity / inattention) is relevant to the field, especially since more individualised treatment options for patients with severe SUDs are highly needed.
However, the manuscript bears a number of concerns that make a publication in this journal questionable.
1. Methods are not entirely clear. ASRS a questionnaire with 2 parts: part A (6 items) is for screening purposes and well validated (although some studies reported a low sensitivity at the usual cut off ; eg Luderer et al, Drug and Alcohol Dependence 2019; Bastiaens and Galus, 2017). Part B ( 12 items) has not been formally validated to my knowledge. Additionally, the authors give a brief overview on how they calculated a sub score from the whole 18 items, but at this point the scoring seems quite arbitrary; the reasons for the chosen scoring approach (why dichotomize when you could calculate a sum score? Why dichotomize at different values for different variables?) should be given in more detail in the text, and backed up by references, if possible.
2. What is alcohol use "above daily threshold"? How was it defined? Depending on the definition, the finding of higher alcohol consumption for participants with more ADHD symptoms might be relevant.
3. The study from which this manuscript derived also included longitudinal data eg urine screenings for approximately 6 to 12 months. Data on clinically relevant outcome parameters might also show the relevance of the findings.
4. The references could be updated
Reviewer 2 Report
This is an important, interesting and well-written paper that adds to the literature on comorbidity between SUDs and ADHD. The polysubstance and mental distress findings in particular have direct treatment implications.
My only main comment for improvement is that I think the authors may overstate the significance of the data given the lack of any control group, most notably a group of opioid dependent patients in abstinence-only or buprenorphine treatment. This should be explicitly stated in the limitations beyond the causality cavaet.
On page 2, the authors state that "findings may illuminate whether patients with higher IHI score seek new and novel treatment options due to previous failed treatments or if this reflects a general preference for novelty and new experiences." This is an interesting hypothesis for future research but it is not clear how the results from this paper could support this.
Thank you for this important contribution.
Reviewer 3 Report
Summary
In the first part of the article, Karlsson et al. aimed to estimate the prevalence of impulsivity, hyperactivity and inattention (IHI) in patients with substance use disorder (SUD) seeking treatment with extended release naltrexone (XR-NTX). To get this information they screened patients using the Adult ADHD Self-Report-Scale (ASRS). In this study, four out of ten (n = 66) patients scored above the clinical cut-off of the ASRS.
Secondly, they compared patients with IHI and patiens without IHI in terms of mental and physical health and substance use by using specific questionnaires (e.g. MINI, ASRS-18, ASRS-6, H-SCL-25, Europ-ASI). In their study higher levels of IHI were significantly associated with longer history of frequent amphetamine use, lifetime violent behaviour, current alcohol use, and higher mental distress.
Finally, they concluded that screening for IHI, in patients with substance use disorder, enables clinicians to better meet the needs of these patients, improve qualitiy of care and therefore decrease risk for discontinuation of treatment.
Strengths
- One major strength of this study is the fact that it is an open label multi-center study, carried out at five urban addiction clinics across Southern Norway. Hence, not only one but serveral clinics were involved in the process of collecting patients.
- The treatment they offer (Vivitrol) is a relatively new substance (phase IV study) in the treatment of substance use disorder. Therefore this study is in different ways innovative and interesting.
- Another strength of this study is that trained staff at the participating hospitals provided detailed information and carried out screening procedures.
- They used the ASRS-18 for detailed information and the ASRS-6 to compare study redults with other studies.
- The conclusion and the disscusion section is well-written and comprehensible.
Limitations
There are several limitations, already mentioned by the authors in their manuscript.
- The ASRS only assesses how often symptoms occured over the past 6 months. However, ADHD is a disease which begins during childhood and can persist into adulthood. Hence, a positive ASRS score in samples with SUD cannot be interpreted as a likely ADHD diagnosis. Nonetheless it is correct that they use the term levels of impulsivity, hyperactivity and inattention and not the term ADHD to describe symptoms of the participating patients.
- Symptoms such as problems remembering appointments and the feeling of restlessness or fidgety can also be associated with withdrawal symptoms or other psychiatric disease. In general, it is hard to tell which symptom is associated with which disease. IHI can be symptom of many different conditions, not only ADHD. It would be of high interest to know, how many of the patients with IHI and SUD would be confirmed as cases with ADHD when examined by a specialized clinician.
- The questionnaires are subjective data, neuropsychological tests for more objetive data were not performed.
- The group size of 162 patients is relatively small. The larger the group size, the greater the statistical power.
Questions
Normally patients with ADHD have worse living conditions (higher divorce rate, imprisoned, poorer education) than in the general population. What is the authors explanation for the only minor differences between groups regarding sociodemographic facts?
Maybe the patients with IHI in this study are the patients with severe symptoms of SUD. Maybe this has nothing to do with ADHD. Did the authors control for withdrawal symptoms possibly mimicking ADHD symptoms in patients with more severe symptoms of SUD?